# Quantification of Water Sources in a Coastal Gold Mine through an End-Member Mixing Analysis Combining Multivariate Statistical Methods

**Guowei Liu [1,2,3], Fengshan Ma [1,2,*], Gang Liu [4], Jie Guo [1,2], Xueliang Duan [1,2] and Hongyu Gu [5]**

[1]   Key Laboratory of Shale Gas and Geoengineering, Institute of Geology and Geophysics, Chinese Academy of Sciences, Beijing 100029, China; liuguowei115@mails.ucas.edu.cn (G.L.); guojie@mail.iggcas.ac.cn (J.G.); 13051876966@163.com (X.D.)

[2]   Innovation Academy of Earth Science, Chinese Academy of Sciences, Beijing 100029, China

[3]   College of Earth and Planetary, University of Chinese Academy of Sciences, Beijing 100049, China

[4]   Xi'an Center of China Geological Survey, Xi'an 710054, Shanxi, China; liugang_iggcas@163.com

[5]   Chengdu Center, China Geological Survey, Chengdu 610081, Sichuan, China; xy0909040129@126.com

\*   Correspondence: fsma@mail.iggcas.ac.cn

**Abstract:** Mixing calculations have been widely applied to identify sources of groundwater recharge, but these calculations have assumed that the concentrations of end-members are well known. However, the end-members of water remain unclear and are not easily available in practical applications. To better determine end-members and mixing ratios, an end-member mixing analysis combining multivariate statistical methods was used on a large, complex water chemistry dataset collected from the Shashandao gold mine in China. Multivariate statistical methods, including principal component analysis (PCA) and hierarchical cluster analysis (HCA), were applied to determine the specific end-members (these two methods verified each other). On the basis of the identified end-members, a maximum likelihood method was then used to estimate the mixing ratios of the water sources. The combined method proposed in this study can help to identify more accurate end-members and deal with uncertainty in end-member concentrations, and it can also adjust the concentrations until the optimal mixing ratios for the calculation are obtained. This method can be a powerful tool for groundwater management and in predicting water inrush in mining operations.

**Keywords:** mixing calculation; ratios; principal component analysis; hierarchical cluster analysis; end-member mixing analysis; maximum likelihood method

## 1. Introduction

In coastal gold mines, it is essential to accurately predict a mixture's water sources and mixing ratios. Mixing calculations have been applied in several scientific fields, including hydrology [1] and other geosciences such as ecology [2], sedimentology [3], and geology [4,5]. In hydrology, mixing calculations have often been used to estimate the mixing ratios of a sample (based on an assumption that a mixture has two or more specific end-members) [6]. A practical approach to end-member mixing analysis (EMMA) in combination with a principal component analysis (PCA) has also been used to analyze real chemical data. In that study, a PCA was used to reduce the dimensions of the datasets and then find extreme component sources [7–9]. Laaksoharju et al. have created a multivariate mixing and mass balance (M3) method using complex hydrochemical data to trace the original water sources of groundwater and compute its mixing proportions and mass balances [10]. This method uses a multivariate analysis called PCA to summarize the datasets and then uses PCA plot mixing calculations

to calculate the mixing proportions of the groundwater samples. The final step in an M3 calculation is to evaluate the deviation from the ideal mixing model.

A PCA consists of a linear transformation of original raw variables into new variables, where each new variable is a linear combination of the old variables. A PCA requires that new variables explain as much of the total explained variance as possible [9,11,12]. Hierarchical cluster analyses (HCAs) are the most widely applied clustering techniques in earth sciences [13–15]. An HCA joins the most similar observations and then connects the next most similar observations. The process repeats until the number of similar observations is reduced to one. Moreover, the levels of similarity at which observations are merged are used to construct a dendrogram in which it is easy to find which observations belong to which clusters [16]. Moore et al. [17] and Liu et al. [18] have simulated potential sources that affect discharge using a PCA based on hydrochemistry. Colby [19] has used a cluster analysis in conjunction with standard geological and hydrogeological analyses to delineate physical and chemical components in hydrogeological units. A principal component factor analysis and a κ-means cluster analysis were applied to groundwater chemistry to better understand groundwater flow and evolution [20]. Combining a PCA with a cluster analysis could divide an entire hydrochemical system into several hydrochemical regimes [21]. To quantify karst aquifer discharge components during storm events, Doctor performed a PCA and an end-member mixing analysis [9] to estimate mixing ratios through the use of major ion chemistry (with the stable isotopes of water and dissolved inorganic carbon). PCA, HCA, and EMMA—multivariate statistical methods based on hydrochemical data—have also been used in some studies to identify water sources, end-members, and their quantity: EMMA in particular has been used to characterize groundwater flow and estimate the mixing proportions of unknown end-members [11,12]. To assess heavy mental contaminant in water and monitor surface water quality, PCA and cluster analysis was used [22,23]. To study groundwater flow, groundwater mode, groundwater evolution and groundwater interaction, PCA combined cluster analysis was also used [24]. To distinguish groundwater sources and calculate ratios of groundwater sources, PCA combined end-member mixing method was applied [11]. Some scholars used an end-member mixing analysis to identify geochemical process, estimate recharge from recirculated groundwater, and calculate mixing ratios [25–27]. Multivariate statistical analysis was applied to evaluate groundwater geochemical evolution and aquifer connectivity and predict sources of water inrush [28,29]. However, for many studies, it might not be possible to find actual end-members. For example, water sources to an aquifer might be already distributed infiltration and undergo hydrochemical reaction. The number of end-members might also be hard to get for many applications. Moreover, a few studies used end-member mixing method based on the maximize likelihood function to calculate mixing ratios of water sources.

The method in this paper has three aspects which are different from previous studies: (1) it used PCA combing HCA to identify end-members (specific water samples and its number) rather than through PCA or extreme value of concentrations; (2) using identified end-members, an end-member mixing method based on the maximize likelihood function was applied to calculate mixing ratios. In process of calculation, the method could acknowledge uncertainty of end-members as well as adjust concentrations of end-members until obtain optimal mixing ratios; (3) establishing three-dimensional geological model to verify the correction of method. In hydrology, mixing calculation is viewed as preliminary step in process of building conceptual model of predicting mine water inrush.

## 2. Study Area

### 2.1. Geological Structures

The Sanshandao gold mine (Figure 1) is located on the east coast of China adjacent to the Bohai Sea. In terms of tectonics, the study area is located at the junction between the Tan-Lu fault and the Jiyang depression [30]. Most of the Sanshandao-Cangshang fracture was buried by Quaternary sediment, and now there is only a partial outcrop. The fracture, which stretches into the Bohai sea, starts at the

town of Sanshandao in the northeast and ends at the village of Panjiawuzi, but there is also an outcrop near Furong island. The length of the fracture is about 12 km, and the width is 50–200 m, forming an "S" shape on the surface. The overall strike of the fracture (facing a southeasterly direction with a dip angle of 45°–75°) is about 40°. This fracture controls three gold deposits: the Sanshandao gold mine, Xinli gold mine, and Cangshang gold mine. The Sanshandao gold mine—the study area—is located in the northeastern part of the Sanshan-Cangshang fracture. In the study area, there are three controlling fractures, which we call F1, F2, and F3 (Figure 1).

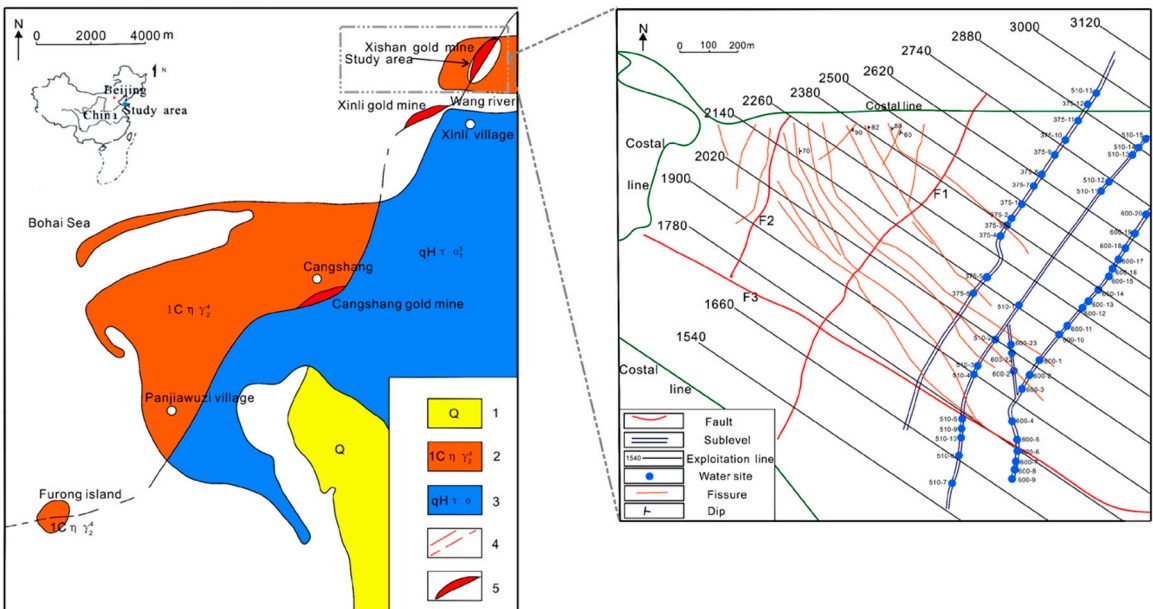

**Figure 1.** Sketch of the study area and sampling location and corresponding IDs of the water samples. 1: Quaternary rock; 2: the Cuizhao unit of the Linglong superunit; 3: the Luanjiazhai unit of the Malianzhuang superunit; 4: a Fault; 5: the gold mine.

The F1 fault, which has a length of about 1300 m and a width of 70–185 m, controls the ore of the mining area. The ore body is located at the footwall of F1, which is inclined to the southeast and has an average dip angle of 46°. Along the strike and tendency, especially the tendency, the F1 fault is smooth and wavy, and the change in strike is obviously greater than the change in tendency. There is a 50–500-mm-thick black gouge and a fracture zone, which has a width of about 70–185 m. According to local geophysical exploration data, the fractured rock descends to a depth of −600 m. It can be inferred that the F1 fracture is extremely large, and the fault gouge and fault fracture rock lead to the F1 fault being stable and having a strong water barrier.

The F2 fault stretches into the Bohai Sea in the northeast and intersects with the F3 fault in the south. Its strike is 280°, and its dip angle is between 60° and 80°. It is a compression-torsion fault. The width of the fracture zone is 5–10 m, which varies a lot due to the behavior of the rock mass along the strike. Some sections are clearly crushed and fractured, and a fault gouge has developed. In other sections, there are shear joint zones and segmental water conductivity. A geophysical exploration showed that the F2 fault has very low-resistance characteristics, indicating that the fault itself has perfect water conductivity. However, the F2 fault is far from the ore mining area, so it has little effect on the stability of the mining tunnel and on water inrush disasters. The F3 fault is located in the southern part of the mining area. It is a regional fault that traverses the mining area. It cuts through the ore-bearing alteration zone and extends into the sea in the northeast. Its strike is 300°–310°, and its dip angle is 80°–90°. A geophysical investigation showed that the F3 fault zone widens with depth, increasing from 3 to 30 m above −250 m.

The F3 fault zone begins to narrow below −300 m, but descends below −600 m. The F3 fault has had multiple periods of activity. The lamprophyre veins along the fault zone are filled, but have been broken, indicating that a tectonic space formed before they were filled and that there was tectonic activity after cementing. Due to the profound depth of the F3 fracture, the fracture zone formed by later tectonic activity is not cemented, and the fill is poor. Therefore, the F3 fault zone is rich in groundwater storage. Moreover, because the F3 fault was displaced and cut into by the F1 fault, it stretches into the Bohai Sea in a northwesterly direction. All of this has led to the F3 fracture becoming a potential water-conducting tunnel connecting the Sanshandao gold mine with seawater in the northwest. Therefore, an in-depth discussion of the hydrogeological engineering characteristics of the fault and disaster prediction and prevention is of great significance to the exploitation of this mine.

*2.2. Hydrogeology*

The groundwater system in the mine area can be divided into a Quaternary aquifer and aquiclude (I), a fractured water-bearing zone in the upper wall of the F1 fault (II), a fractured water-bearing zone in the footwall of the F1 fault (III), and a fractured water-bearing zone in the F3 fault (IV).

In the Quaternary aquifer and aquiclude (I), the Quaternary sediment is widely distributed in the mining area. In terms of lithology, the Quaternary sediment can be separated into four parts from top to bottom: the first aquifer ($I_1$), the first aquiclude ($I_2$), the second aquifer ($I_3$), and the second aquiclude ($I_4$). The first aquifer ($I_1$) consists of both medium and course sand, and its thickness ranges from 3.5 to 17.29 m. Near the surface, in the range 0–2 m, the sand is yellowish-brown, and the particle size is uniform and pure. Organic matter increases with increased depth. The pore water and water-bearing content vary greatly because of the existence of the gouge, with the hydraulic conductivity ranging from 5.35 to 15.27 m/day. Moreover, atmospheric precipitation and seawater recharge are the main renewable groundwater resources for the first aquifer ($I_1$), and the total dissolved solid content ranges from a value of 0.21 to 25.95 g/L. The first aquiclude ($I_2$) is located below the first aquifer ($I_1$), with its burry depth ranging from 5.5 to 9.0 m. Its thickness does not clearly change, and its general thickness is 7–8 m. The lithology is mainly sandy clay. This layer has a relatively poor water barrier. The second aquifer ($I_3$) is located below the first aquiclude and is not continuous. Its thickness gradually increases from north to south. Its average thickness is 3–4 m, with a maximum thickness of 11.9 m. It is a confined aquifer and receives a recharge from seawater and the first aquifer ($I_1$). It mainly consists of medium sand, course sand, and gravel. The second aquiclude ($I_4$), which has a burry depth ranging from 7.8 to 25.5 m, is located above the weathered crust. This layer has a 3–5-m thickness: it is stable, and its lithology includes red clay. All of these characteristics make it a good water barrier.

In the fractured water-bearing zone in the upper wall of the F1 fault (II), weathered fissures and tectonic fissures containing confined groundwater were developed in the granite and metamorphic rocks on the upper wall of the F1 fault. The top and bottom of the water-bearing zone include biotite granite and Jiaodong metamorphic rocks. The rock is dense and hard. The lower F1 fault gouge and mylonite belt are a good barrier to the groundwater entering the pit. In fact, all of these characteristics make this area a relatively good water barrier.

In the fractured water-bearing zone in the footwall of the F1 fault (III), there are many fault structures, which have become the main location for groundwater activity. The footwall of the F1 fault has a wide distribution of fractures, ranging from the beginning of the F1 fault in the east to the F2 fault in the west. In the south, fractures have developed up to the 1540 exploration line. The hydrogeological characteristics are different due to varying tectonic development characteristics and groundwater recharge conditions.

In terms of the fractured water-bearing zone of F3 (IV), the F3 fault is a large regional fault that traverses through the middle of the mining area and cuts across the F1 fault. The F3 fault has experienced multiple periods of activity, especially recently; therefore, the rock in the belt of the fault is completely broken, and the breccia is not cemented. This provides space for the storage and

transportation of groundwater and the infiltration of seawater. Due to the large width of the fault belt, this area is prone to various hydrogeological and engineering geological disasters.

## 3. Methods

### 3.1. Water Sampling

The water samples were taken from the −375-m sublevel, −510-m sublevel, and −600-m sublevel of the Xishan gold mine. In August of every year from 2009 to 2016 (except for 2010), two bottles of 600-ml water samples were collected at each water site. The water samples were packed in plastic bottles and sealed. To ensure quality, the water samples were stored in a refrigerator at 4 °C prior to analysis. $K^+$, $Na^-$, $Ca^{2+}$, $Mg^{2+}$, $Cl^-$, $SO_4{}^{2-}$, $HCO_3{}^-$, pH, electrical conductivity (EC), and total dissolved solids (TDSes) were measured at the Institute of Geology, China Earthquake Administration. All water samples were measured by a DIONEX-500 (Thermo Fisher Scientific, Waltham, USA) ion chromatograph, and the $CHO_3{}^-$ content in the water samples was tested using an METROHM$^{TM}$ (Metrohm, Herisau, Switzerland) The cations, anions, TDSes, pH, and EC were 0.01 mg/L, 0.1 mg/L, 0.01 mg/L, 0.01, and 0.1μs/cm, respectively. The testing standards of the cations and anions had detection limits of 0.05 and 0.1 mg/L, respectively, and were based on a DZ/T0064.28-1993 and a DZ/Too64.51-1993 [31] (Luo et al., 2017). More water samples were used for an environmental isotope analysis ($\delta^{18}O$ and $\delta^2H$). The measurements were made at the Laboratory for Stable Isotope Geochemistry, Institute of Geology and Geophysics (Beijing, China). Isotope analysis values with an analytical precision defined by two-sigma uncertainties of 0.2% for $\delta^{18}O$ and 2% for $\delta^2H$ were measured by an MT-253 mass spectrometer. The values of $\delta^{18}O$ and $\delta^2H$, which were expressed in ‰, were in accordance with standard mean ocean water (SMOW) values [32]. The chemistry datasets are included in Table S1 of the Supplementary Materials. In Table S1, it can be seen that in most water sites, there were several water samples collected, but for some water sites, there was only one water sample. This was due to seasonal rainfall and fissure water decreasing or drying out, but in addition, due to the exploitation of the gold mine, more water sites were gradually added as tunnels were excavated. For example, water site 375-2 was sampled in 2009 but dried out afterwards, and water site 375-9 was added in 2012.

### 3.2. Principal Component Analysis (PCA)

Principal component analysis is a statistical technique consisting of the linear transformation of original variables into new variables (principal components), where the principal components are a linearization of the old variables [16]. The principal components align with the greatest variance in the multivariate datasets. PCA is used to reduce the complexity of large multivariate datasets and then elucidate underlying patterns that are not obviously present within the datasets. The first principal component explains as much as possible of the total explained variance of the variables. The second principal component explains as much as possible of the residual total explained variance of the variables, and so on. Further, the first principal component accounts for more total explained variance than any other principal component does, and all principal components are mutually orthogonal. The term "principal component score" refers to the values of new variables that are projected onto the principal axis. By plotting the sites sampled using the principal component score as the plotting position, sample relations and groupings became evident, and then extreme-value points—potential end-members—could be identified. The PCA was performed on SPSS 25.0 software (SPSS Inc., New York, NY, USA).

### 3.3. Hierarchical Cluster Analysis (HCA)

Cluster analysis is a branch of multivariate techniques whose primary purpose is to cluster objects based on properties they possess. According to predetermined selection criteria, cluster analysis divides objects into a few clusters so that objects in the same cluster have the most similar characteristics [12].

HCA is the most widely used cluster analysis technique in hydrogeology, and it can identify the initial relationship between any one sample and the entire dataset, typically visually illustrating the relationship with a dendrogram [33,34]. In this study, HCA was performed on standardized hydrochemistry datasets using Ward's linkage rule, which generates different groups based on an analysis of variance and then groups all nonresiduals into separate clusters, where each sample in a cluster is more similar to the other samples in the same cluster than to samples in other clusters. The Euclidean distance [35,36] is used in HCA as a measure of similarity between all samples. Here, we conducted an HCA using SPSS 25.0 software to help identify the number of end-members applied to the end-member mixing analysis model.

*3.4. End-Member Mixing Analysis (EMMA)*

With the assumption that a groundwater sample is composed of two or more water sources, end-members that represent the characteristics of sources of groundwater inflow into a system and their proportions can be calculated using end-member mixing analysis [1,9,11,12]. In this study, we thought that since the end-members could not be sampled, they could be obtained using PCA based on hydrochemical datasets. Here, we used a proposed algorithm that is described in detail in the maximum likelihood framework provided by Carrera [1]. Equation (1) allows for any numbers of end-members, estimating the proportions of end-members:

$$c_{i,j} = \sum_{k=1}^{n} f_{i,k} E_{j,k} + \alpha_{p,s} \tag{1}$$

where $c_{i,j}$ is the concentration of parameter $j$ for water site $i$, $f_{i,k}$ is the fraction (or mixing ratios) of end-members $k$ that is associated with water site $i$, $E_{j,k}$ is the end-member concentration for variable $j$ and end-member $k$, and $\alpha_{p,s}$ is the error. This algorithm was programed using MIX software, which was created by Carrera. This algorithm mainly comprises four steps (Figure 2): (1) assuming that the concentrations of end-members are known, the mixing proportions are identified using conventional least squares (initialization); (2) given the initial mixing proportions, the maximum likelihood function is used to estimate the expected values of the samples and the end-member concentrations; (3) given the expected values of the samples and the end-member concentrations from (2), the maximum likelihood method is used again to calculate the mixing proportions; and (4) the algorithm does not end until there is convergence and no residual reduction between step 2 and step 3. In the traditional EMMA method, there is a limitation: when water sources flow into a system, hydrochemical evolution is neglected. However, because it acknowledges the uncertainty of end-member concentrations, the method presented in this paper has two advantages. First, it can improve the estimation of actual end-member concentrations. Second, and most importantly, it can improve the accuracy of the calculated mixing ratios.

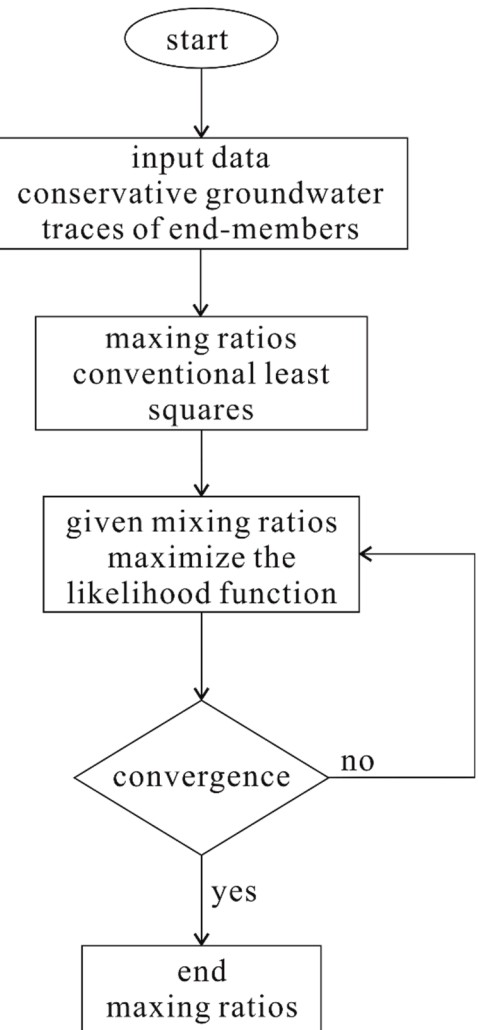

**Figure 2.** Framework of calculating mixing ratios with end-member mixing analysis (EMMA).

## 4. Application and Results

In most geological research, the magnitudes and dimensions of variables are important. Standardization tends to enhance the influence of variables that have small variances and diminish the influence of variables that have large variances. The variables used in the PCA and HCA were standardized according to the formula [9]

### 4.1. Scenario 1: −375-m Sublevel

A principal component analysis (PCA) combined with a hierarchical cluster analysis (HCA) was performed on the variables, including $K^+$, $Na^+$, $Ca^{2+}$, $Mg^{2+}$, $Cl^-$, $SO_4^{2+}$, $HCO_3^-$, pH, EC, and the TDSes of 38 water samples collected from the −375-m sublevel. The results, including the water sites plotted from the PCA, are demonstrated in Figure 3. Principal components 1, 2 and 3 accounted for 52.7%, 22.3%, and 9.4%, respectively, of total explained variance. These three principal components explained 84.4% of the hydrochemistry information. In Figure 3, two extreme-value points (end-members) can clearly be seen (freshwater and 375-13-1), as can three potential extreme-value points (approximate end-members) (seawater, 375-6-1, and 375-5-5) [7].

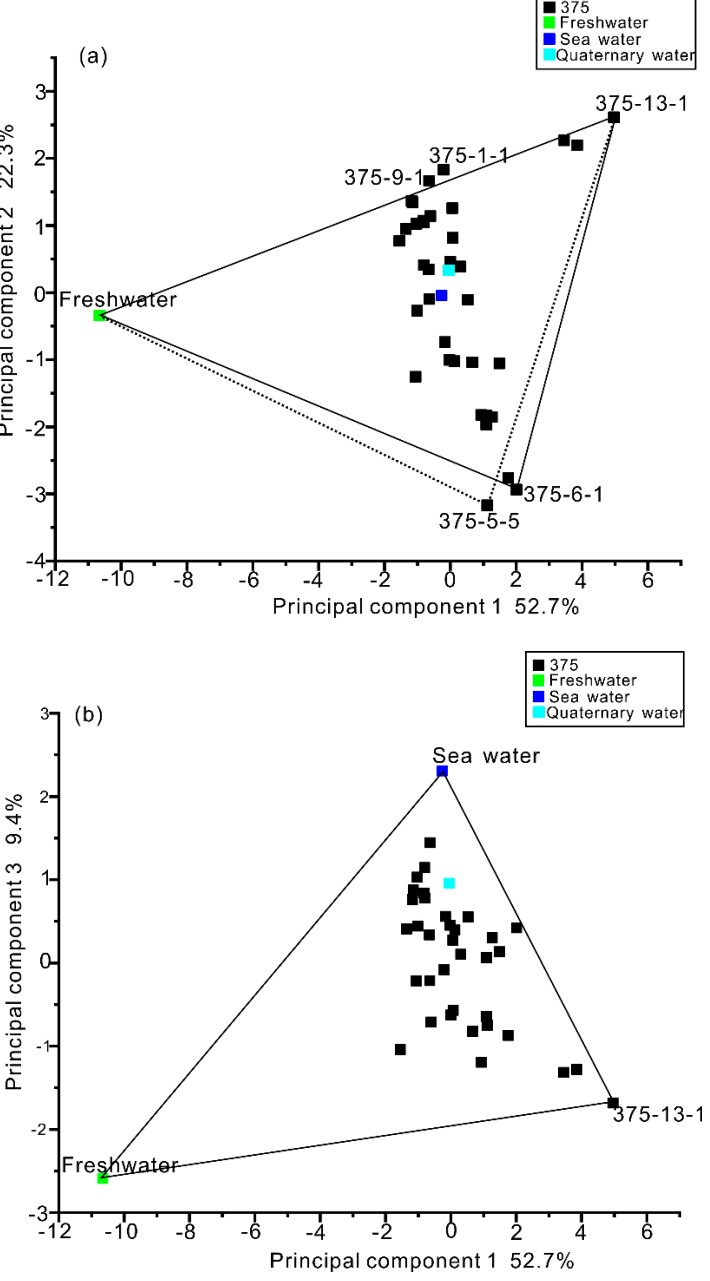

**Figure 3.** Results of the principal component analysis (PCA) for water samples collected from the −375-m sublevel. (**a**) Plot scatters of scores of the principal component 1 and 2. (**b**) Plot scatters of scores of the principal component 1 and 3. These three principal components account for 84.4% of the total explained variance.

The three potential end-members were then estimated using an HCA, which was performed on raw data from the −375-m water samples. The HCA detected similar groups among the water samples and then produced a dendrogram (Figure 4) grouping 38 water samples into four clusters. The criterion for identifying the four groups was linkage distance, defined as $D_{link}/D_{max}$ [33], which is a quotient of linked distance at a special point divided by the maximum distance. Combining the results from the PCA and HCA, it could be seen that there were four end-members at the −375-m sublevel. Freshwater and 375-13-1 were two of these end-members. We used prior knowledge and Figure 3b to determine that seawater was another end-member. We later chose not to include water samples 375-5-5 and 375-6-1 as end-members. With 375-5-5, there were two water samples located significantly

outside the polygon that consisted of freshwater, 375-13-1, and 375-5-5. Besides, the distance between 375-5-5 to the polygon was relatively far. As with 375-5-5, the samples 375-9-1 and 375-1-1 were outside the polygon. However, they were near the line with freshwater and 375-13-1 and had a good fit with this line. This demonstrated that these two water samples were mainly composed of freshwater and 375-13-1. The results (Table 1) from analyzing the mixing ratios of 375-13-1, which were calculated afterwards, also proved this. For example, for 379-1-1, freshwater sources accounted for 31.5% and the 375-13-4 water source accounted for 59.3%, and these two end-members accounted for 90.8% of all water sources. For 375-9-1, freshwater water sources accounted for 33.1% and the 375-13-1 water source accounted for 57.0%, and these two end-members accounted for 90.1% of all water sources.

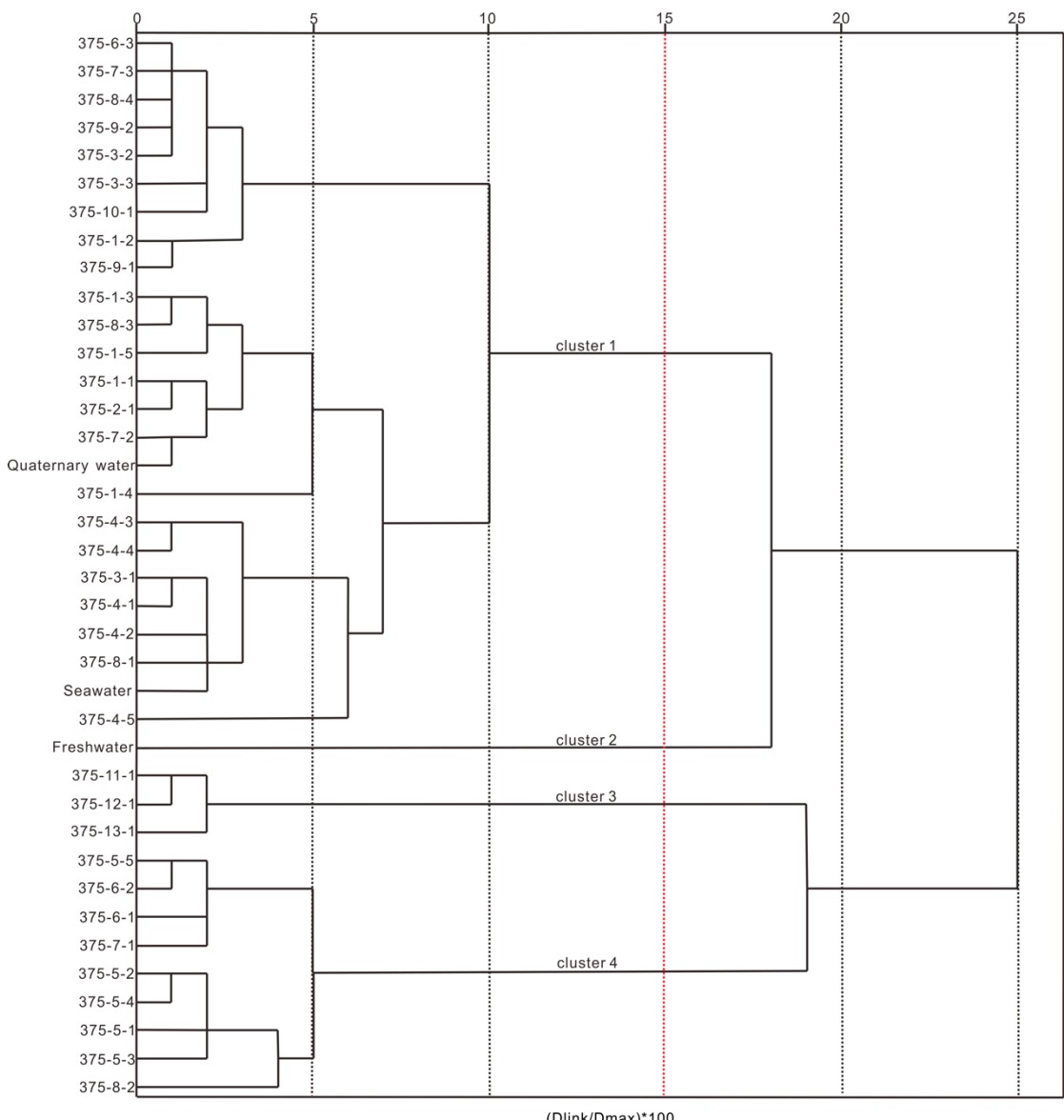

**Figure 4.** Dendrogram of the hierarchical clustering analysis (HCA) of water samples collected from the −375-m sublevel in the Xishan gold mine.

**Table 1.** Results of calculated proportions of each end-member in a sample for every sample of -375 m sublevel.

| Location | End-Members of −375m Sublevel | | | |
|---|---|---|---|---|
| | Freshwater | Seawater | 375-6-1 | 375-13-1 |
| 375-1-1 | 0.315 | 0.092 | 0 | 0.593 |
| 375-1-2 | 0.06 | 0.542 | 0.361 | 0.037 |
| 375-1-3 | 0.122 | 0.392 | 0.325 | 0.162 |
| 375-1-4 | 0.366 | 0.023 | 0 | 0.611 |
| 375-1-5 | 0.37 | 0.045 | 0.05 | 0.535 |
| 375-2-1 | 0.336 | 0.024 | 0.034 | 0.607 |
| 375-3-1 | 0.027 | 0.702 | 0.271 | 0 |
| 375-3-2 | 0.459 | 0 | 0.001 | 0.54 |
| 375-3-3 | 0.431 | 0 | 0 | 0.569 |
| 375-4-1 | 0.318 | 0.102 | 0.239 | 0.34 |
| 375-4-2 | 0.442 | 0 | 0.026 | 0.532 |
| 375-4-3 | 0.113 | 0.371 | 0.423 | 0.092 |
| 375-4-4 | 0.362 | 0.058 | 0.02 | 0.56 |
| 375-4-5 | 0.363 | 0.036 | 0.033 | 0.568 |
| 375-5-1 | 0.161 | 0.162 | 0.19 | 0.487 |
| 375-5-2 | 0.11 | 0.281 | 0.318 | 0.291 |
| 375-5-3 | 0.222 | 0.107 | 0.322 | 0.349 |
| 375-5-4 | 0.306 | 0.005 | 0.002 | 0.687 |
| 375-5-5 | 0.061 | 0.311 | 0.374 | 0.253 |
| 375-6-2 | 0.037 | 0.308 | 0.35 | 0.305 |
| 375-6-3 | 0.096 | 0.615 | 0.22 | 0.069 |
| 375-7-1 | 0.105 | 0.213 | 0.391 | 0.291 |
| 375-7-2 | 0.399 | 0.035 | 0.053 | 0.513 |
| 375-7-3 | 0.35 | 0.085 | 0.137 | 0.429 |
| 375-8-1 | 0.364 | 0 | 0.076 | 0.56 |
| 375-8-2 | 0.203 | 0.17 | 0.047 | 0.58 |
| 375-8-3 | 0.191 | 0.322 | 0.107 | 0.38 |
| 375-8-4 | 0.393 | 0.097 | 0 | 0.51 |
| 375-9-1 | 0.337 | 0.059 | 0.084 | 0.52 |
| 375-9-2 | 0.089 | 0.619 | 0.221 | 0.072 |
| 375-10-1 | 0.129 | 0.261 | 0.311 | 0.298 |
| 375-11-1 | 0 | 0 | 0 | 1 |
| 375-12-1 | 0.025 | 0.138 | 0.074 | 0.763 |
| Quaternary water | 0.406 | 0.025 | 0.004 | 0.565 |

*4.2. Scenario 2: The −510-m Sublevel*

Using the same procedure as in scenario 1, PCA and HCA were performed on datasets of samples from the −510-m sublevel (demonstrated in Figures 5 and 6, respectively). Principal component 1, principal component 2, and principal component 3 represented 64.8%, 13.2%, and 11.0% of total explained variance, respectively. These three principal components perfectly expressed the hydrochemical information of the datasets. We chose four extreme-value points (freshwater, seawater, 510-8-1, and 510-11-1) as end-members. As is shown in Figure 5, 510-7-1 was another extreme value point. We did not choose it as an end-member for two reasons: the HCA demonstrated that the water samples from the −510-m sublevel were divided into four clusters (Figure 6): if we had chosen 510-7-1 as an end-member, there would have been many water samples located outside of the polygon consisting of freshwater, seawater, and 510-7-1 or consisting of freshwater, 510-8-1, and 510-7-1. Thus, the 510-7-1 water sample did not represent a water source and could not be an end member.

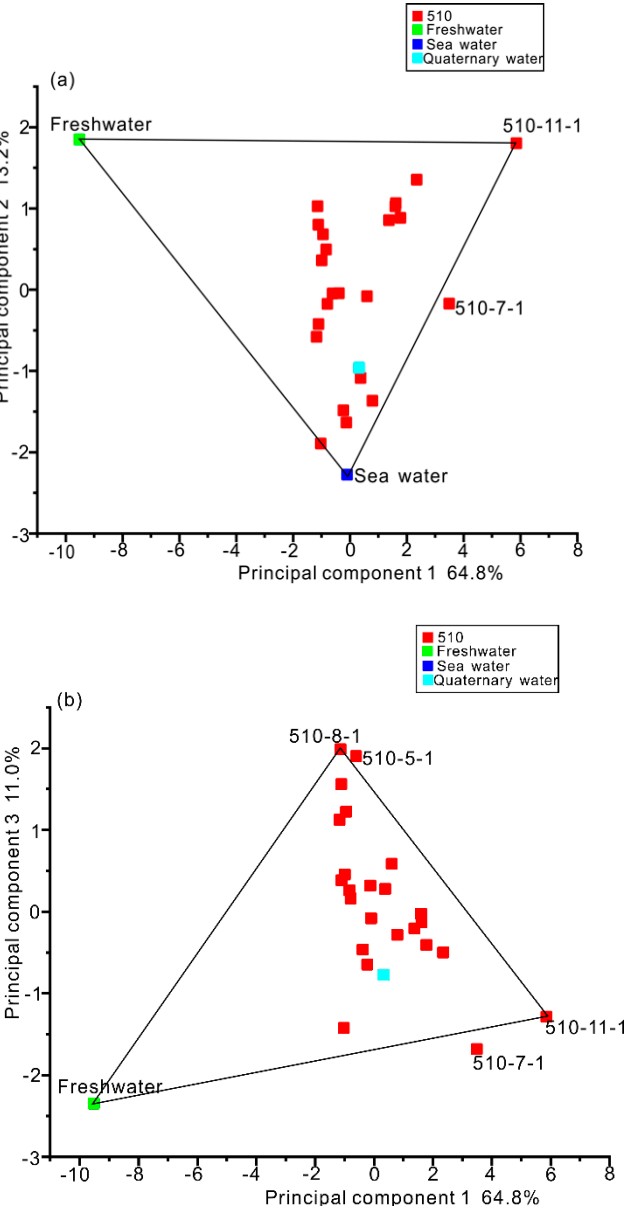

**Figure 5.** Results of the principal component analysis for water samples collected from the −510-m sublevel. (**a**) Plot scatters of scores of the principal component 1 and 2. (**b**) Plot scatters of scores of the principal component 1 and 3. These three principal components account for 89.0% of the total explained variance

### 4.3. Scenario 3: −600-m Sublevel

The results from the PCA are shown as water samples plotted on the first three principal components, which represented 93.7% of total explained variance in the data from the −600-m sublevel (Figure 7). Principal components 1, 2, and, 3 represented 62.9%, 21.8%, and 9.0%, respectively, of total explained variance. The HCA showed that the water samples from the −600-m sublevel were divided into four clusters. When the PCA and HCA were combined (Figure 8), freshwater, seawater, 600-8-1, and 600-11-1 were obtained as the four end-members. The sample 600-3-1 was located on the line between seawater and 600-8-1 and was near 600-8-1. This demonstrated that 600-3-1 could be represented by 600-8-1 to a large extent. Besides, in Figure 7b, it can clearly be seen that 600-3-1 was in the polygon consisting of freshwater, seawater, 600-8-1, and 600-11-1. Therefore, 600-3-1 was excluded and could not be used as an end-member.

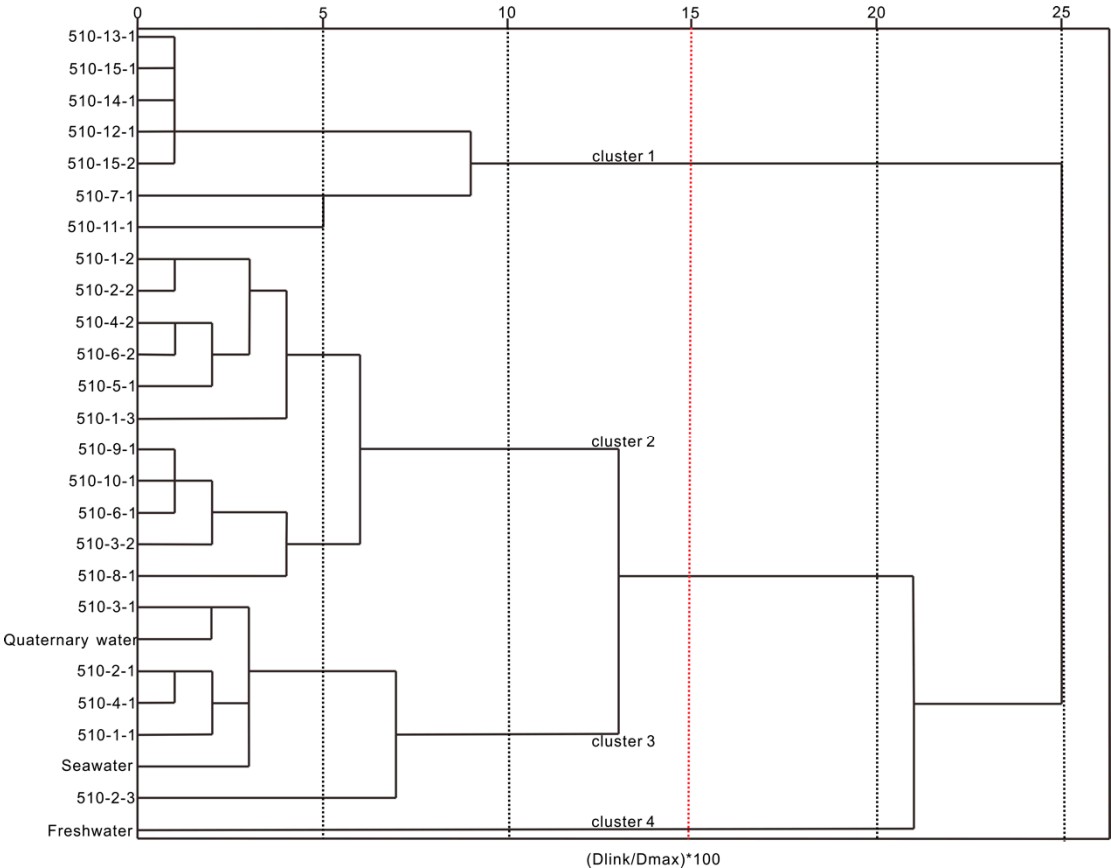

**Figure 6.** Dendrogram of the hierarchical clustering analysis for water samples collected from the −510-m sublevel in the Xishan gold mine.

*4.4. Mixing Ratios of Water Sources*

An EMMA method based on the theory of maximum likelihood was used to calculate the mixing ratios of the Xishan gold mine. Here, 38 water samples from the −375-m sublevel, 26 water samples from the −510-m sublevel, and 43 water samples from the −600-m sublevel were collected, for which the following three parameters were used, respectively: $\delta^{18}O$, $\delta^2H$, and $Cl^-$. Using the above results, freshwater, seawater, 375-6-1, and 375-13-1 were selected as end-members for the −375-m sublevel; freshwater, seawater, 510-8-1, and 510-11-1 were selected as end-members for the −510-m sublevel; and freshwater, seawater, 600-8-1, and 600-11-1 were selected as end-members for the −600-m sublevel. The calculated proportions of the water samples from the Xishan gold mine are shown in Tables 1–3. According to Tables 1–3, the authors calculated the average proportions (Figure 9) of the samples at each water site. In Figure 9, it can clearly be seen that with depth, the ratios of the seawater generally increased. However, an opposite phenomenon happened with freshwater. At the −600-m sublevel, the contribution of seawater was approximately 40%, and in some water sites, the ratio was 50% or more, with freshwater contributing little to the four end-members.

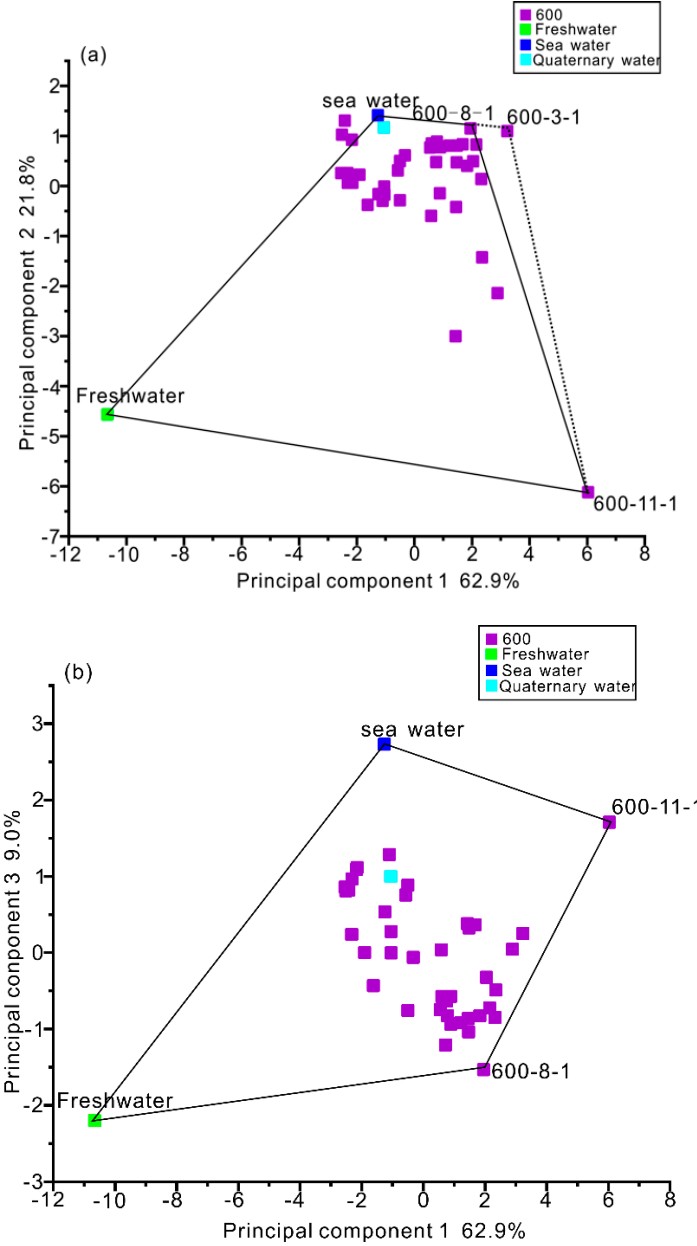

**Figure 7.** Results of the principal component analysis for the water samples collected from the −600-m sublevel. (**a**) Plot scatters of scores of the principal component 1 and 2. (**b**) Plot scatters of scores of the principal component 1 and 3. These three principal components account for 93.7% of the total explained variance.

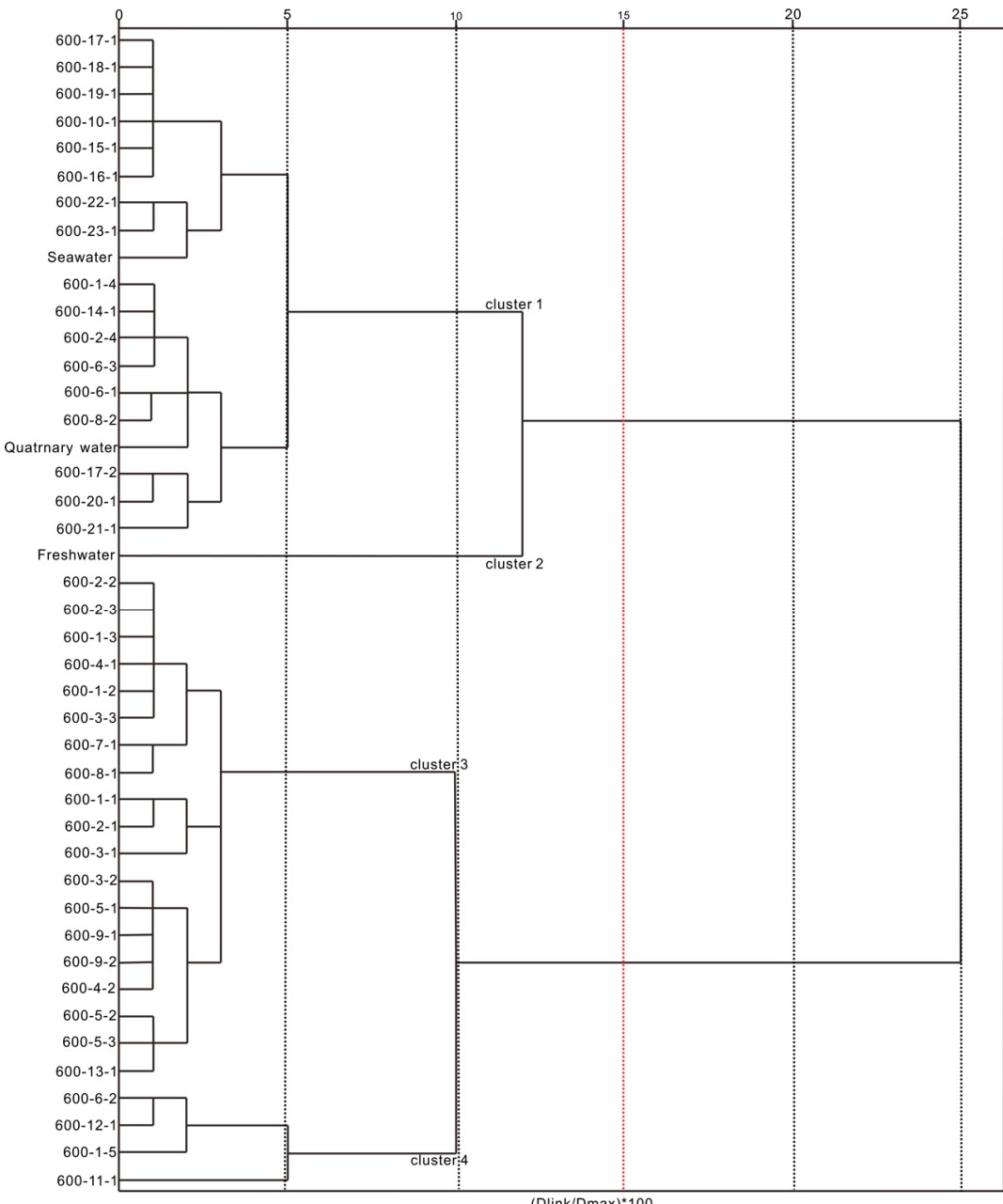

**Figure 8.** Dendrogram of the hierarchical clustering analysis for water samples collected from the −600-m sublevel in the Xishan gold mine.

**Table 2.** Results of calculated proportions of each end-member in a sample for every sample of −510 m sublevel.

| Location | End-Members of −510 m Sublevel | | | |
|---|---|---|---|---|
| | Freshwater | Seawater | 510-8-1 | 510-11-1 |
| 510-1-1 | 0.178 | 0.029 | 0.595 | 0.197 |
| 510-1-2 | 0 | 0.057 | 0.88 | 0.063 |
| 510-1-3 | 0.005 | 0.045 | 0.8 | 0.149 |
| 510-2-1 | 0.002 | 0.049 | 0.803 | 0.416 |
| 510-2-2 | 0 | 0.077 | 0.839 | 0.084 |
| 510-2-3 | 0 | 0.031 | 0.935 | 0.034 |
| 510-3-1 | 0.749 | 0.234 | 0.017 | 0 |
| 510-3-2 | 0.008 | 0.03 | 0.829 | 0.133 |

**Table 2.** *Cont*.

| Location | End-Members of −510 m Sublevel | | | |
| --- | --- | --- | --- | --- |
| | **Freshwater** | **Seawater** | **510-8-1** | **510-11-1** |
| 510-4-1 | 0.245 | 0.056 | 0.535 | 0.164 |
| 510-4-2 | 0 | 0.067 | 0.859 | 0.074 |
| 510-5-1 | 0.254 | 0.042 | 0.634 | 0.07 |
| 510-6-1 | 0.083 | 0.033 | 0.824 | 0.061 |
| 510-6-2 | 0 | 0.063 | 0.832 | 0.105 |
| 510-7-1 | 0.006 | 0 | 0.341 | 0.653 |
| 510-9-1 | 0.278 | 0.067 | 0.61 | 0.045 |
| 510-10-1 | 0.455 | 0.071 | 0.382 | 0.092 |
| 510-12-1 | 0.001 | 0.013 | 0.556 | 0.43 |
| 510-13-1 | 0 | 0.028 | 0.521 | 0.451 |
| 510-15-1 | 0.001 | 0.037 | 0.5 | 0.461 |
| Quaternary water | 0.401 | 0.061 | 0.368 | 0.17 |

**Table 3.** Results of calculated proportions of each end-member in a sample for every sample of -600 m sublevel.

| Location | End-Members of −600 m Sublevel | | | |
| --- | --- | --- | --- | --- |
| | **Freshwater** | **Seawater** | **600-8-1** | **600-11-1** |
| 600-1-1 | 0.117 | 0.063 | 0.821 | 0 |
| 600-1-2 | 0 | 0.312 | 0 | 0.688 |
| 600-1-3 | 0 | 0.32 | 0 | 0.68 |
| 600-1-4 | 0.229 | 0.232 | 0.539 | 0 |
| 600-1-5 | 0 | 0.296 | 0 | 0.704 |
| 600-2-1 | 0.094 | 0.112 | 0.782 | 0.013 |
| 600-2-2 | 0 | 0.337 | 0.002 | 0.661 |
| 600-2-3 | 0.001 | 0.232 | 0.514 | 0.253 |
| 600-2-4 | 0.112 | 0.315 | 0.573 | 0 |
| 600-3-1 | 0.053 | 0 | 0.946 | 0 |
| 600-3-2 | 0 | 0.203 | 0 | 0.797 |
| 600-3-3 | 0.03 | 0.132 | 0.838 | 0 |
| 600-4-1 | 0.002 | 0.317 | 0 | 0.68 |
| 600-4-2 | 0 | 0.119 | 0.777 | 0.105 |
| 600-5-1 | 0 | 0.132 | 0.462 | 0.406 |
| 600-5-2 | 0.086 | 0.167 | 0.748 | 0 |
| 600-5-3 | 0.385 | 0 | 0.615 | 0 |
| 600-6-1 | 0.037 | 0.262 | 0.701 | 0 |
| 600-6-2 | 0.208 | 0.002 | 0.79 | 0 |
| 600-6-3 | 0 | 0.469 | 0 | 0.53 |
| 600-7-1 | 0.03 | 0.182 | 0.773 | 0.016 |
| 600-8-2 | 0.066 | 0.266 | 0.669 | 0 |
| 600-9-1 | 0.026 | 0.116 | 0.84 | 0.017 |
| 600-9-2 | 0.035 | 0.151 | 0.814 | 0 |
| 600-10-1 | 0.083 | 0.371 | 0.545 | 0 |
| 600-12-1 | 0 | 0.18 | 0 | 0.82 |
| 600-13-1 | 0 | 0.34 | 0 | 0.66 |
| 600-14-1 | 0 | 0.419 | 0 | 0.58 |
| 600-15-1 | 0 | 0.495 | 0 | 0.504 |
| 600-16-1 | 0 | 0.509 | 0 | 0.49 |
| 600-17-1 | 0 | 0.52 | 0 | 0.48 |
| 600-17-2 | 0.095 | 0.35 | 0.001 | 0.554 |
| 600-18-1 | 0.003 | 0.524 | 0.002 | 0.471 |
| 600-19-1 | 0.003 | 0.537 | 0.001 | 0.458 |
| 600-20-1 | 0.075 | 0.271 | 0.462 | 0.192 |
| 600-21-1 | 0.099 | 0.269 | 0.224 | 0.408 |
| 600-22-1 | 0.072 | 0.372 | 0.547 | 0.009 |
| 600-23-1 | 0 | 0.473 | 0.169 | 0.358 |
| Quaternary water | 0.186 | 0.287 | 0.527 | 0 |

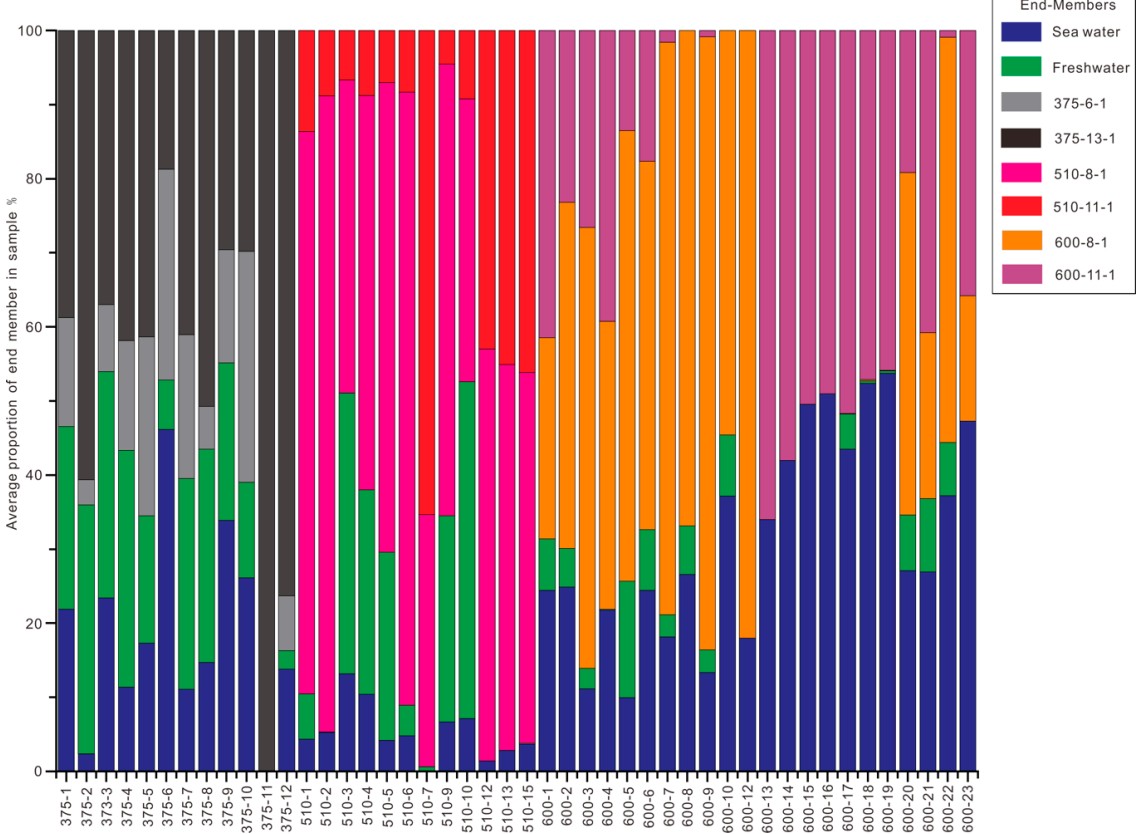

**Figure 9.** The average proportions of the water samples for each water site at the −375-m sublevel, −510-m sublevel, and −600-m sublevel.

## 5. Discussion

On the basis of local hydrogeological conditions, previous studies, and the results from scenarios 1, 2 and 3, seawater and freshwater could be identified as the two main water sources (end-members) in the study area. In these three scenarios, there was another possible water sample: quaternary water. Many scholars studying the region have used quaternary water as a water source. However, in this research, quaternary water was not revealed to be a water source. Depending on the hydrogeological conditions, quaternary water is mainly recharged from seawater and atmospheric precipitation. It also has the characteristics of seawater and freshwater. Importantly, the quaternary water sample here did not have extreme values and was relatively far from the extreme-value points on the scatterplots of the principal component scores. This all demonstrated that quaternary water was not suitable for selection as a water source (end-member).

### 5.1. Choice of End-Members in the PCA and HCA Models

There are two types of groundwater in the rock fractures of the Xishan gold mine: Ca-rich water and Mg-rich water [37]. The chemical compositions of these two types of water evolved from quaternary water, freshwater, and sea water. Mining operations and artificial drainage not only accelerate the groundwater flow rate and then enrich the quaternary water, freshwater, and sea water, but also cause an ion exchange reaction that removes the calcium in the potassium feldspar and the magnesium in the hornblende. In Figure 10, it can be seen that there were some water samples (375-13-1, 510-11-1, and 600-11-1) that contained Mg-extreme values and some water samples (375-6-1, 375-6-2, 510-11-1, and 600-11-1) that contained Ca-extreme values. By comparing Figure 10 to the results of the PCA and HCA, it was easily found that the two results confirmed each other. It was reasonable to select seawater, freshwater, 375-6-1, and 375-11-1 at the −375-m sublevel; seawater, freshwater, and

510-11-1 at the −510-m sublevel; and seawater, freshwater, 600-8-1, and 600-11-1 at the −600-m sublevel (510-11-1 and 600-11-1 were rich in calcium and magnesium). Here, the water sample 510-8-1 was unique and was not consistent with the PCA and HCA analyses. Principal component 3 accounted for 11.0% of the information from the original variables, with significant loading of pH and $HCO_3^-$. Moreover, 510-8-1 contained the highest value of $HCO_3^-$ of all the collected water samples (Figure 11). Thus, it was also reasonable to select 510-8-1 as an end-member at the −510-m sublevel. The reasons for the high content of $HCO_3^-$ were that volatile components, including water, carbon dioxide, and methane, exist in the Sanshandao gold mine; bereseite and lamprophyre are the main rocks containing carbonate minerals; and Quaternary water and freshwater containing $CO_2$ flow into the fractures [38].

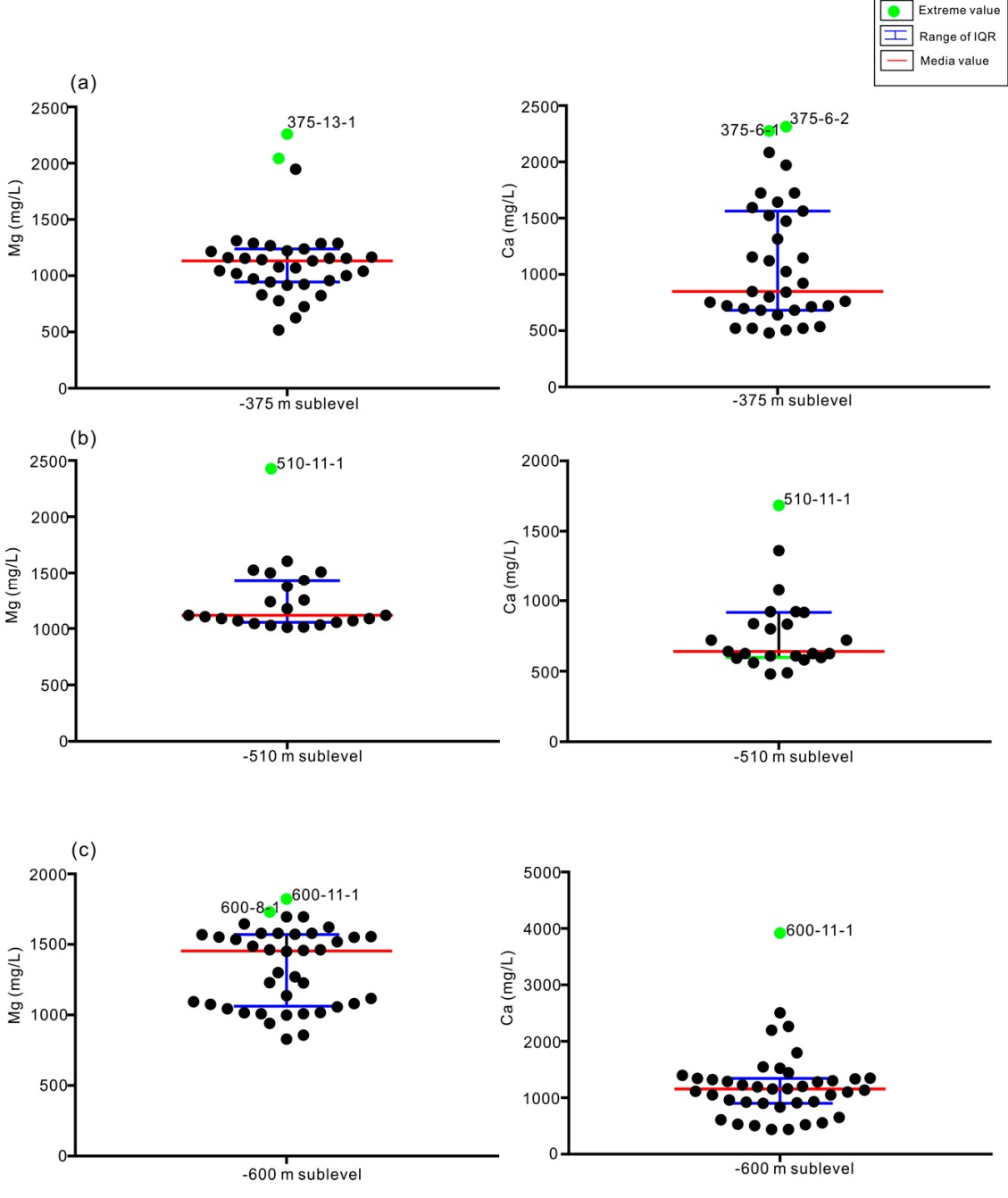

**Figure 10.** Scatterplots of the concentration values of the variables: Mg and Ca in the water samples collected from the main sublevels: (**a**) the −375-m sublevel, (**b**) −510-m sublevel, and (**c**) −600-m sublevel.

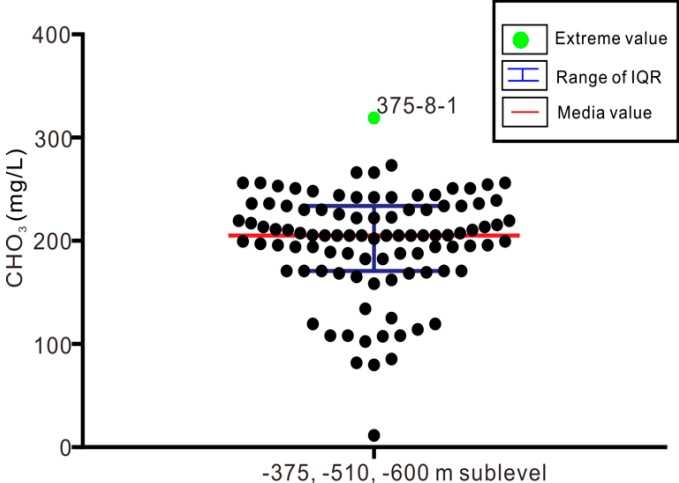

**Figure 11.** Scatterplots of the concentration values of the variables: $CHO_3$ in the water samples collected from the −375-m sublevel, −510-m sublevel, and −600-m sublevel.

*5.2. Choice of Conservative Groundwater Tracers in the EMMA Model*

The stable isotopes of water ($\delta^{18}O$ and $\delta^2H$) are usually used as conservative groundwater tracers to estimate the proportions of water mixes [9,11,12]. In our research in the tunnels of the Xishan gold mine, four end-members were identified. According to Equation (2), there should be three tracers used to obtain mix sources (end-members). Here, the authors chose $Cl^-$ as another tracer to estimate the mixing ratios. Ten parameters were used to estimate the mix proportions of the water samples. Table 4 shows the principal component loadings of all of the original variables. The table demonstrates that $Cl^-$ had extremely high loadings in PC1s, which explains most of the hydrochemical information in the datasets collected from the three sublevels (0.978, 0.967, and 0.990). Tables 5–7 show a Person correlation matrix of the ten variables. They demonstrate that $Cl^-$ had a strong, positive correlation with five variables ($Na^+$, $Ca^{2+}$, $Mg^{2+}$, $SO_4^{2+}$, EC, and TDSes) and a strong, negative correlation with one variable (pH). $Cl^-$ is nonvolatile, hydrologically mobile, and chemically inert, and it is assumed to be affected only by groundwater mixing [11,12,20]. All of this accounts for why $Cl^-$ was chosen as another conservative groundwater tracer.

**Table 4.** Table of component weightings, principal component Eigenvalues, and the variance of the first three principal components at the −375-m sublevel, −510-m sublevel, and −600-m sublevel.

| | −375-m Sublevel | | | −510-m Sublevel | | | −600-m Sublevel | | |
|---|---|---|---|---|---|---|---|---|---|
| | PC1 | PC2 | PC3 | PC1 | PC2 | PC3 | PC1 | PC2 | PC3 |
| $K^+$ | 0.403 | −0.308 | 0.837 | 0.370 | −0.807 | 0.377 | 0.134 | 0.498 | 0.851 |
| $Na^+$ | 0.986 | 0.029 | −0.043 | 0.987 | 0.058 | 0.057 | 0.966 | 0.200 | −0.055 |
| $Ca^{2+}$ | 0.630 | −0.580 | −0.381 | 0.855 | 0.328 | −0.250 | 0.768 | −0.596 | 0.020 |
| $Mg^{2+}$ | 0.630 | 0.735 | −0.057 | 0.962 | 0.160 | −0.017 | 0.894 | 0.260 | −0.208 |
| a $Cl^-$ | **0.978** | 0.088 | −0.115 | **0.967** | 0.147 | −0.006 | **0.990** | 0.023 | −0.056 |
| $SO_4^{2+}$ | 0.830 | 0.110 | 0.112 | 0.942 | −0.104 | 0.165 | 0.705 | 0.660 | −0.053 |
| $HCO_3^-$ | −0.128 | 0.932 | −0.013 | 0.047 | 0.600 | 0.530 | −0.459 | 0.724 | −0.334 |
| pH | −0.166 | 0.605 | 0.166 | −0.297 | 0.155 | 0.757 | −0.583 | 0.695 | −0.104 |
| EC | 0.822 | −0.029 | 0.153 | 0.904 | −0.332 | 0.095 | 0.967 | 0.156 | 0.032 |
| TDS | 0.987 | 0.065 | −0.100 | 0.994 | 0.081 | −0.002 | 0.993 | 0.065 | −0.044 |
| Eigenvalue | 5.269 | 2.231 | 0.937 | 6.483 | 1.320 | 1.098 | 6.288 | 2.181 | 0.902 |
| PEVCPEV | 52.7 | 22.3 | 9.4 | 64.830 | 13.198 | 10.978 | 62.878 | 21.811 | 9.02 |
| | 52.7 | 75.0 | 84.4 | 64.830 | 78.028 | 89.006 | 62.878 | 84.689 | 93.710 |

Note: PC = Principal component; PEV = Percentage of explained variance; CPEV = Cumulative percentage of explained variance. a Significant loadings of $Cl^-$ (>0.5) are in bold.

**Table 5.** Pearson correlation coefficients for 10 variables from water samples collected from the −375-m sublevel. EC: electrical conductivity; TDSes: total dissolved solids.

| | −375-m Sublevel | | | | | | | | | |
|---|---|---|---|---|---|---|---|---|---|---|
| | $K^+$ | $Na^+$ | $Ca^{2+}$ | $Mg^{2+}$ | $Cl^-$ | $SO_4^{2+}$ | $HCO_3^-$ | pH | EC | TDSes |
| $K^+$ | 1.000 | | | | | | | | | |
| $Na^+$ | 0.359 | 1.000 | | | | | | | | |
| $Ca^{2+}$ | 0.141 | 0.605 | 1.000 | | | | | | | |
| $Mg^{2+}$ | −0.004 | 0.645 | −0.049 | 1.000 | | | | | | |
| $Cl^-$ | 0.292 | 0.985 | 0.628 | 0.689 | 1.000 | | | | | |
| $SO_4^{2+}$ | 0.357 | 0.778 | 0.398 | 0.571 | 0.764 | 1.000 | | | | |
| $HCO_3^-$ | −0.312 | −0.099 | −0.627 | 0.616 | −0.042 | −0.038 | 1.000 | | | |
| pH | −0.145 | −0.162 | −0.325 | 0.212 | −0.100 | −0.055 | 0.408 | 1.000 | | |
| EC | 0.399 | 0.792 | 0.436 | 0.467 | 0.750 | 0.577 | −0.137 | −0.128 | 1.000 | |
| TDS | 0.309 | 0.990 | 0.635 | 0.674 | 0.998 | 0.795 | −0.066 | −0.120 | 0.758 | 1.000 |

**Table 6.** Pearson correlation coefficients for 10 variables from the water samples collected from the −600-m sublevel.

| | −600-m Sublevel | | | | | | | | | |
|---|---|---|---|---|---|---|---|---|---|---|
| | $K^+$ | $Na^+$ | $Ca^{2+}$ | $Mg^{2+}$ | $Cl^-$ | $SO_4^{2+}$ | $HCO_3^-$ | pH | EC | TDSes |
| $K^+$ | 1.000 | | | | | | | | | |
| $Na^+$ | 0.185 | 1.000 | | | | | | | | |
| $Ca^{2+}$ | −0.169 | 0.632 | 1.000 | | | | | | | |
| $Mg^{2+}$ | 0.071 | 0.902 | 0.491 | 1.000 | | | | | | |
| $Cl^-$ | 0.104 | 0.971 | 0.764 | 0.893 | 1.000 | | | | | |
| $SO_4^{2+}$ | 0.361 | 0.812 | 0.134 | 0.785 | 0.697 | 1.000 | | | | |
| $HCO_3^-$ | 0.046 | −0.264 | −0.761 | −0.215 | −0.402 | 0.141 | 1.000 | | | |
| pH | 0.164 | −0.421 | −0.830 | −0.285 | −0.543 | 0.037 | 0.691 | 1.000 | | |
| EC | 0.233 | 0.945 | 0.639 | 0.901 | 0.947 | 0.783 | −0.329 | −0.484 | 1.000 | |
| TDS | 0.131 | 0.984 | 0.739 | 0.903 | 0.995 | 0.735 | −0.387 | −0.514 | 0.957 | 1.000 |

**Table 7.** Pearson correlation coefficients for 10 variables from the water samples collected from the −510-m sublevel.

| | −510-m Sublevel | | | | | | | | | |
|---|---|---|---|---|---|---|---|---|---|---|
| | $K^+$ | $Na^+$ | $Ca^{2+}$ | $Mg^{2+}$ | $Cl^-$ | $SO_4^{2+}$ | $HCO_3^-$ | pH | EC | TDSes |
| $K^+$ | 1.000 | | | | | | | | | |
| $Na^+$ | 0.334 | 1.000 | | | | | | | | |
| $Ca^{2+}$ | −0.045 | 0.819 | 1.000 | | | | | | | |
| $Mg^{2+}$ | 0.213 | 0.966 | 0.865 | 1.000 | | | | | | |
| $Cl^-$ | 0.221 | 0.962 | 0.870 | 0.946 | 1.000 | | | | | |
| $SO_4^{2+}$ | 0.446 | 0.932 | 0.726 | 0.899 | 0.889 | 1.000 | | | | |
| $HCO_3^-$ | −0.132 | 0.111 | 0.034 | 0.087 | 0.088 | −0.016 | 1.000 | | | |
| pH | −0.071 | −0.247 | −0.307 | −0.239 | −0.231 | −0.100 | 0.147 | 1.000 | | |
| EC | 0.653 | 0.869 | 0.644 | 0.772 | 0.817 | 0.872 | −0.027 | −0.311 | 1.000 | |
| TDS | 0.298 | 0.994 | 0.869 | 0.974 | 0.973 | 0.922 | 0.085 | −0.276 | 0.866 | 1.000 |

*5.3. Three-Dimensional Geological Model*

The validity of the EMMA model was further evaluated using fracture distribution law and a three-dimensional geological model. A field investigation of the fracture development characteristics of the area was valuable for studying the seepage channels and mixing ratios. Through years of measurements of the surrounding rock structures at the −375-m, −510-m, and −600-m sublevels, data from 1122 fractures (471, 391, and 260, respectively) were obtained, mainly on inclination, dip, crack spacing, and trace length. Using these data, rose diagrams of crack strike and pole contour maps of

fracture inclination and dip were drawn (Figure 12). As can be seen in Figure 12, the strikes of the fractures in the three sublevels were mainly in a northeasterly direction and secondly in a northwesterly direction. These strikes were 15°–25° and 300°–320°, respectively. Moreover, there were a few fissure zones with an orientation angle between 270° and 280°. In Figure 12, it can be seen that the tendency was stronger in the second and fourth quadrants than in the first and third quadrants. Meanwhile, the fissures oriented toward the north were consistent with the results presented by the strike graph. The dip angles of the multiple sets of fissures were larger, mostly above 60°, and were concentrated between 70° and 90°.

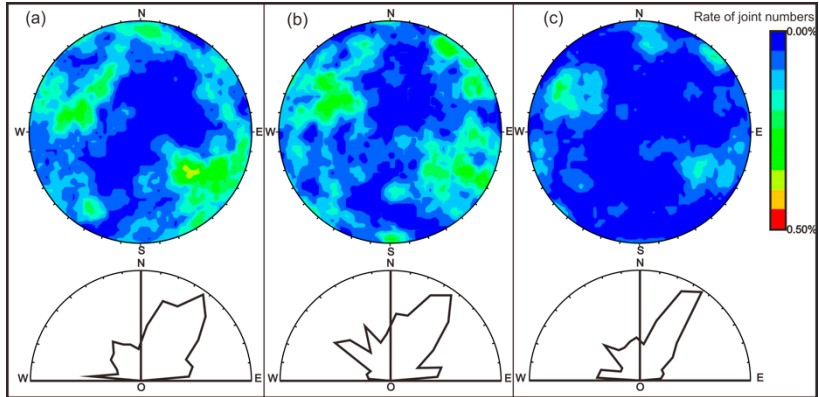

**Figure 12.** Pole contour maps and rose diagrams of the cracks of the main sublevels.

Further, we also analyzed data from 1122 fractures and plot histograms of the frequency distribution of the inclination, dip angles, and crack spacing. In Figure 13, it can be seen that the fractures had discrete characteristics. However, each sublevel had at least two peak points. In a certain range of dip angles, the tendency could be approximated as two normal distributions or log-normal distributions. The fracture tendencies of the −375-m sublevel, −510-m sublevel, and −600-m sublevel all obeyed a log-normal distribution, with average dips of 130° and 297°, 130°and 310°, and 120° and 297°, respectively. Particularly at the −600-m sublevel, there were many fractures with dips of 0°–20°. Because there were too many, these were analyzed separately, and their tendency followed a normal distribution, with a mean dip of 10°.

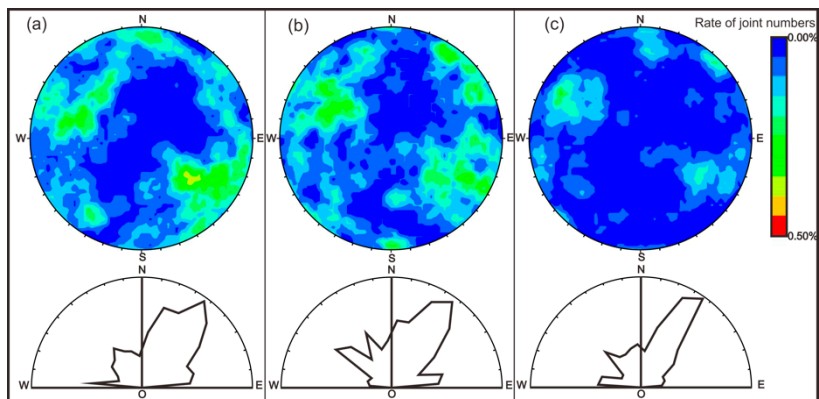

**Figure 13.** Histograms and fitting curves for the frequency distribution of the fracture inclination, collected from the main sublevels of the Xishan gold mine: (**a**) the −375-m sublevel, (**b**) −510-m sublevel, and (**c**) −600-m sublevel.

Since the dip angles of the fissures were generally large and the number of samples was limited, conventional distributions, such as a log-normal distribution or Weibull distribution, were difficult to fit to the fissure data. Thus, a normal distribution was adopted, and the data from the field investigation

were in good agreement with part of the data from the normal distribution curve (Figure 14). The two groups of dip angles at the −375-m sublevel were 85°; the two groups of dip angles at the −510-m sublevel were 87° and 72°; and the three groups of dip angles at the −600-m sublevel were 87°, 77° and 77°.

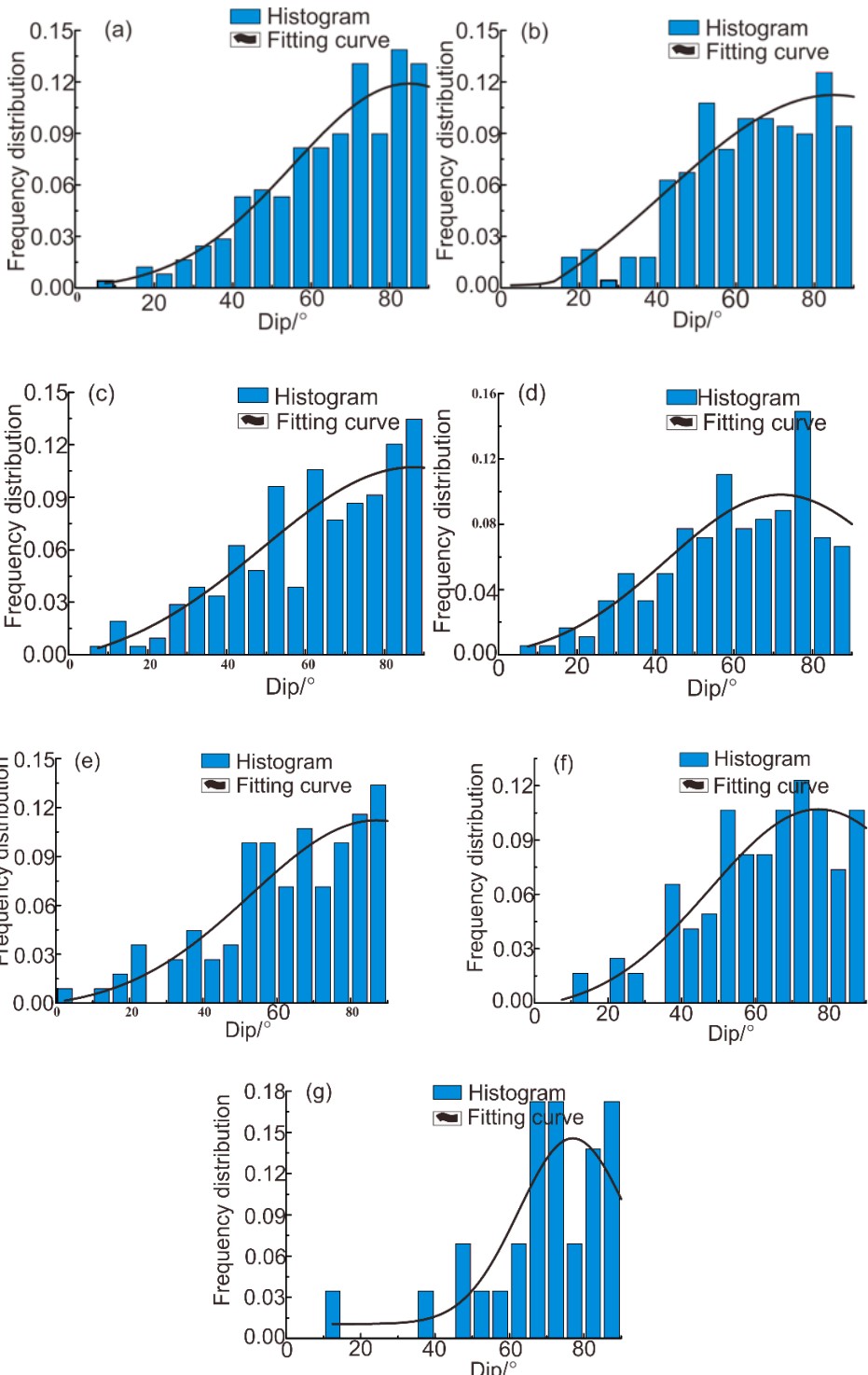

**Figure 14.** Histograms and fitting curves of the frequency distribution of the fracture dip collected from the main sublevels of the Xishan gold mine: (**a**) the first set of fractures at the −375-m sublevel, (**b**) the second set of fractures at the −375-m sublevel, (**c**) the first set of fractures at the −510-m sublevel, (**d**) the second set of fractures at the −510-m sublevel, (**e**) the first set of fractures at the −600-m sublevel, (**f**) the second set of fractures at the −375-m sublevel, (**g**) and the third set of fractures at the −375-m sublevel.

The window method was used to estimate the trace length of the fissure. As is shown in Figure 15, this method is based on relationships that include cutting, intersection, and containment in terms of the boundary of the window. This method has the advantages of flexibility and a large number of samples [39]. After analyzing the dozens of statistical windows in the mine, the average trace length of fissures was found to be 0.8 m. The trace length was assumed to obey a negative exponential distribution [40]. In Figure 16, it can be seen that the crack spacing more or less fit a log-normal distribution. The two groups of crack tracings at the −375-m sublevel were 0.113 m and 0.123 m; the two groups of dip angles at the −510-m sublevel were 0.151 m and 0.135 m; and the three groups of dip angles at the −600-m sublevel were 0.134 m, 0.13 m, and 0.09 m.

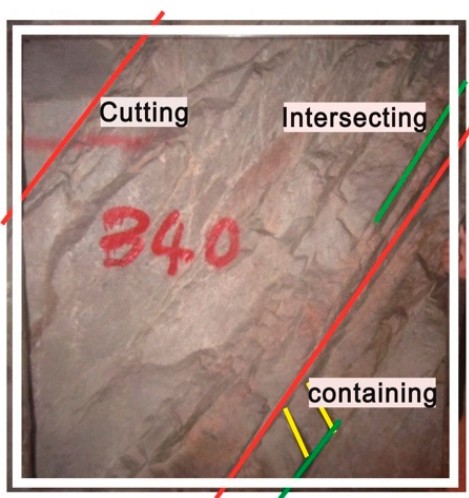

**Figure 15.** Example of the window method in measuring the joint trace.

We used data on the fissure tendency, dip angle, and trace length; a mean and variance estimation; an assumption that the fissure was disc-shaped; and three-dimensional geometric modeling (Figure 17) of the −375-m sublevel, −510-m sublevel, and −600-m sublevel to carry out an analysis using MATLAB software (MathWorks Inc., Natick, USA, 2018). In Figure 17, it can be seen that the fissure had characteristics of high dispersion and a high dip angle ratio and agreed with the actual fissure well. In order to more intuitively display the three faults (F1, F2, and F3) in the mining area, the occurrences (tendency and dip angle) were plotted (Figure 17d). The occurrence of F1 was 127°∠46°, the occurrence of F2 was 282°∠85°, and the strike of F3, which was nearly vertical, was 300°. Combining information from the three sublevels, it could be concluded that the occurrences of the three sets of fractures were 130°∠85°, 300°∠78°, and 10°∠77°, respectively. Comparing the fracture occurrence and fault occurrence, it can be seen that the three groups of fracture occurrence had a strong correlation with the fracture distribution of the mining area: the first group of fractures could be part of F1, the second set of fractures could be closely related to F2, and the third set could be part of the F3 fault. According to the property and position information of the three fault zones, it could be preliminarily determined that the third set of fractures had greater water conductivity than did the first and second sets of fractures.

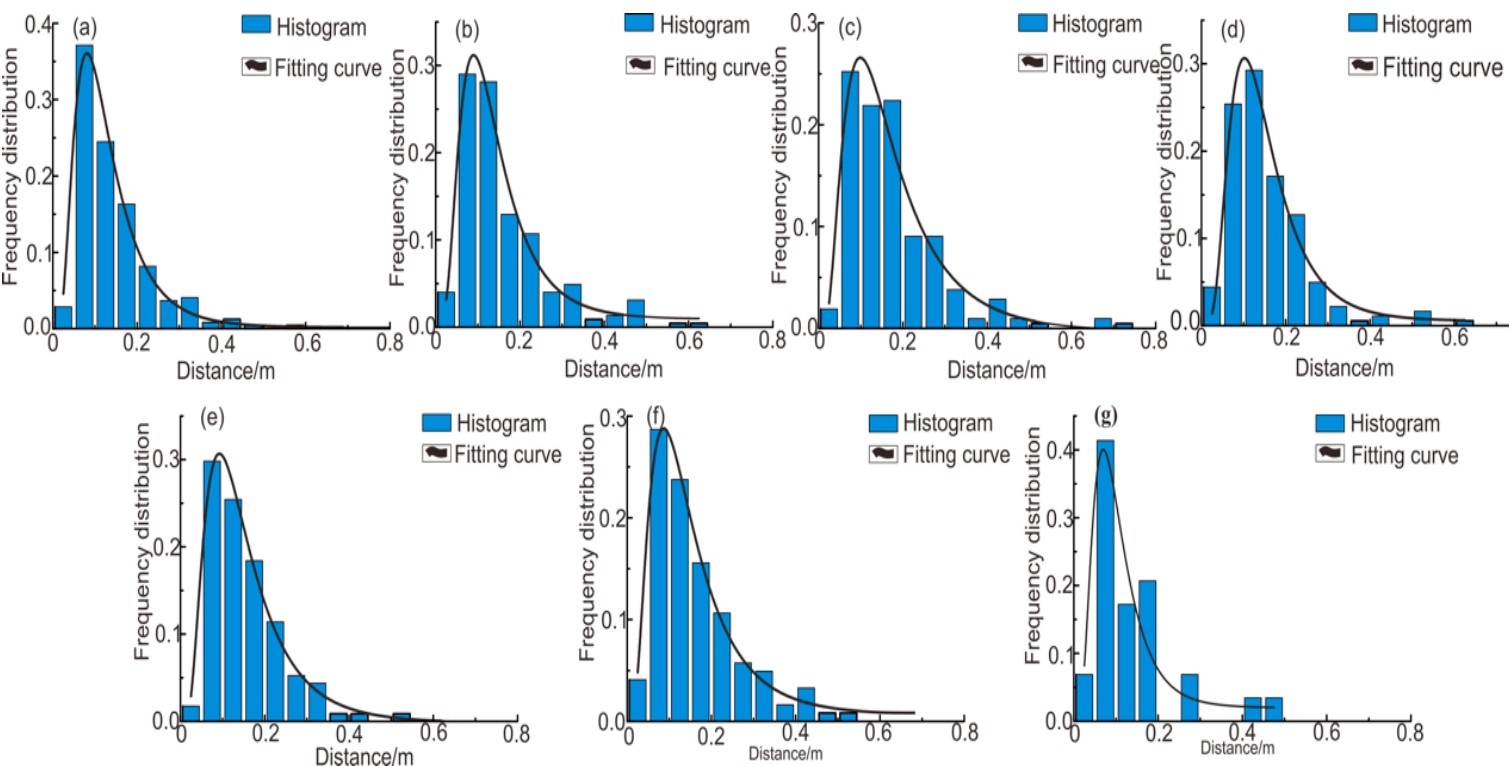

**Figure 16.** Fitting curves of the frequency distribution of fracture spacing collected from the main sublevels at the Xishan gold mine: (**a**) the first set of fractures at the −375-m sublevel, (**b**) the second set of fractures at the −375-m sublevel, (**c**) the first set of fractures at the −510-m sublevel, (**d**) the second set of fractures at the −510-m sublevel, (**e**) the first set of fractures at the −600-m sublevel, (**f**) the second set of fractures at the −375 m sublevel, (**g**) and the third set of fractures at the −375-m sublevel.

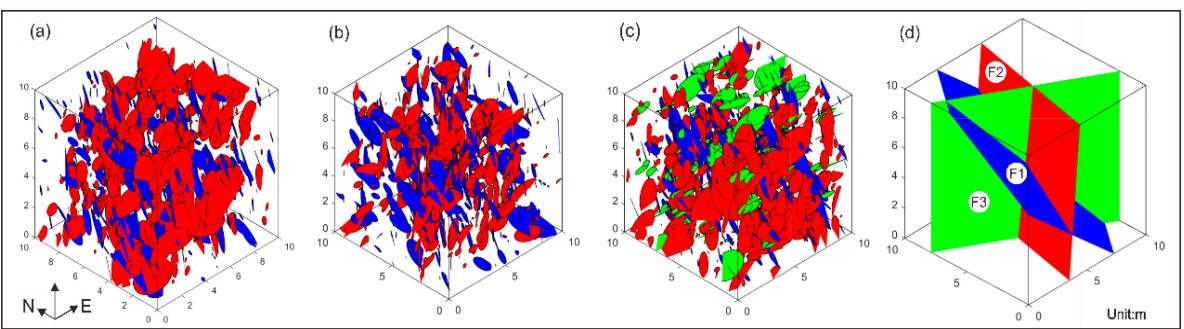

**Figure 17.** Three-dimensional network models of fractures of the three sublevels and a schematic map of the faults in the mining area: (**a**) the −375-m sublevel; (**b**) −510-m sublevel; and (**c**) −600-m sublevel. (**d**) schematic map of the faults in the mining area. Different colors represent different fracture sets and faults.

From discussion above, there are four ways to evaluate the reliability of the chosen methods. 1. It can be seen that the first three principle components (Figure 3) explain most of the water chemistry of the water samples, and four extreme water samples (freshwater, seawater, 375-6-1, and 375-13-1) can be derived from it. It shows that in the −375 m sublevel, all water samples consist of four potential water sources. Using HCA, the water samples (Figure 4) in −375 m sublevel were also divided into four clusters. The results of PCA and HCA agree well. This shows to some extent that the two methods verify each other. Further, by combing these two methods, end-members (the number and specific water samples) can be obtained. For −510 m sublevel and −600 m sublevel, it has the same results of the analysis. 2. Combining with PCA and HCA, the end-members were obtained. According to available researches, the brine of the Sanshandao gold mine mainly contains $Ca^{2+}$ and $Mg^{2+}$. A scatter plot was made of $Ca^{2+}$ and $Mg^{2+}$ ions in the water samples, it was found that the water samples at the extreme value point of $Ca^{2+}$ and $Mg^{2+}$ ions corresponds to the end-members which were obtained. 3. From the PCA results, we selected several special water samples: 375-1-1, 375-9-1, 510-5-1 and 600-3-1. For 375-1-1 and 375-9-1, they are near the line that was composed by freshwater and 375-13-1 and have a good fit with this line. This demonstrated that these two water samples were mainly composed by freshwater and 375-13-1. For 510-5-1, it is near the line that was composed by freshwater and 500-8-1 and near the line which was composed by 510-8-1 and 510-11-1. This demonstrate that 510-5-1 were mainly composed by 510-8-1, freshwater, and 510-11-1. The 600-3-1 was located on the extension line of sea water and 600-8-1 and was near 600-8-1. It demonstrated that 600-3-1 could be represented by 600-8-1 in a large extent. From the EMMA results, it was apparently seen that 375-1-1 was composed by freshwater and 375-13-1, which make up 31.5% and 59.3%, respectively. 375-9-1 was composed by freshwater and 375-13-1, which make up 33.3% and 52.0%, respectively. For 510-5-1, it was composed by freshwater, 510-11-1, and 510-11-1, which make up 25.4%, 63.4%, and 7%, respectively. For 600-3-1, it was mainly composed by 600-8-1 at 94.6%. In summary, the qualitative analysis of PCA and the quantitative analysis of EMMA are consistent. 4. According to Section 5.3, it was found that the third set of fractures of 10°∠77° which belongs to F3 fault has greater water conductivity than the first and second sets of fractures and is widely developed in the −600 m sublevel. The results of EMMA apparently demonstrate that in the −600 m sublevel, the seawater content is abruptly increased. These two results are also consistent.

## 6. Conclusions

Two problems involving two uncertainties were addressed in this paper. One of the problems was the uncertainty of identifying end-members and their quantity within water samples. The other was the uncertainty in terms of the concentrations of different conservative groundwater tracers. The first uncertainty can come from groundwater evolution, groundwater interaction, and ion exchange. The second can come from measurement errors or from spatial and temporal variability within a

water sample. With these two uncertainties, it is not easy to find a mixture representing end-members (water sources). Because of this, we tried to find mixtures that contained a lot of information about end-members to calculate the mixing ratios.

In this paper, we demonstrated a method that combines principal component analysis (PCA), hierarchical cluster analysis (HCA), and end-member mixing analysis (EMMA) to estimate the mixing ratios of a water sample. The first step is the combined use of PCA and HCA. PCA uses the dimensionality reduction of water chemistry datasets to obtain extreme values (potential end-members) and then calculates the number of end-members (according to the HCA), finally obtaining the specific end-members. The second step is to use EMMA to calculate the mixing ratios of the water sample. Using the concentration values of original end-members as original values, EMMA can obtain new optimum end-members so that the fitted mixing ratios can be calculated. Finally, the results of the mixing ratios were confirmed using fracture distribution law and a three-dimensional geometric model. At the −600-m sublevel, the seawater exceeded 40% in all water sources, and the third set of fractures (10° ∠77°) probably had more water inrush than the other two did.

EMMA combining multivariate statistical methods (PCA and HCA) is vital in hydrology. It is a preliminary step to building conceptual models of the contents of water sources of groundwater and where water sources originate. It can provide evidence of and support for mine groundwater management, especially in terms of water inrush.

**Supplementary Materials:** The following are available online at http://www.mdpi.com/2073-4441/12/2/580/s1, Table S1: Statistics of the parameters of water samples.

**Author Contributions:** Conceptualization, G.L. (Guowei Liu); Formal analysis, J.G.; Funding acquisition, F.M.; Investigation, G.L. (Guowei Liu), G.L. (Gang Liu) and H.G.; Supervision, F.M. and X.D.; Validation, G.L. (Gang Liu); Writing—original draft, G.L. (Guowei Liu); Writing—review & editing, G.L. (Guowei Liu). All authors have read and agreed to the published version of the manuscript.

**Funding:** This work has been conducted in the framework of t the National Science Foundation of China (Grant nos. 41831293, 41907174) and the National Key Research Projects of China (2016YFC0402802).

**Conflicts of Interest:** The Authors declare no conflicts of interest.

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
