# Peer review of "Quantification of Water Sources in a Coastal Gold Mine through an End-Member Mixing Analysis Combining Multivariate Statistical Methods"

_water, doi:10.3390/w12020580_

Round 1

Reviewer 1 Report

In this paper, the authors used PCA and HCA together with the end-member mixing analysis method to help identify sources of groundwater recharge and their mixing ratios. A case study for the Shashandao gold mine in China was used to demonstrate the effectiveness of the proposed methodological framework. In general, the paper is well written and organized. However, I still have a few major comments about their methods:

It is not clear how do you validate the performance of your methods (PCA + HCA). In the abstract, the authors mentioned: “these two methods verify each other”. These two methods are different (for different purposes), how can they verify each other? It is not clear what criteria are used here to evaluate the models’ or methods’ performance. The authors should clearly describe them and show the evaluation results for your methods. How do your models/methods perform in comparison to other existing methods (if any)? In other words, the authors should justify why your methods are required and whether or not your methods are comparable to others. It is not clear how uncertainties in end-member concentrations and the mixing ratios are dealt with. It is suggested to include a flow chart to summarize all the methods/data used here and how they worked or interacted with one another in step-by-step fashion.

Reviewer 2 Report

Please, see the enclosed file.

Round 2

Reviewer 1 Report

The authors have addressed my comments properly. I think it is now acceptable for publication.

Reviewer 2 Report

I am glad that the Authors followed all my comments and suggestions to improve their manuscript.

In my opinion, it was substantially improved and it is ready for publication.

Congratulations to the Authors!